# Entropy Optimization in MHD Nanofluid Flow over an Exponential Stretching Sheet

Precious Sibanda [1], Mohammed Almakki [2], Zachariah Mburu [3] and Hiranmoy Mondal [4],*

1 School of Mathematics, Statistics and Computer Science, University of KwaZulu-Natal, Pietermaritzburg Private Bag X01, Scottsville, Pietermaritzburg 3209, South Africa
2 School of Mathematical and Nature Sciences, American University of Ras Al Khaimah, Ras Al Khaimah P.O. Box 10021, United Arab Emirates
3 Department of Mathematics and Statistics, University of Embu, Embu P.O. Box 6-60100, Kenya
4 Department of Applied Mathematics, Maulana Abul Kalam Azad University of Technology, Kolkata 700064, West Bengal, India
* Correspondence: hiranmoymondal@yahoo.co.in

**Abstract:** We numerically investigate mixed convective heat and mass transport in incompressible nanofluid flow through an exponentially stretching sheet with temperature-dependent viscosity. The fluid flow equations are transformed to a system of non-linear ordinary differential equations using appropriate similarity transformations and solved numerically by using the multi-domain bivariate spectral quasi-linearization technique. The fast convergence of the method is shown by demonstrating that the error is exponentially small for a finite number of iterations. The significance and impact of different fluid parameters are presented and explained. For engineering relevance, the entropy generation number has been calculated for different fluid parameter values.

**Keywords:** entropy generation; variable viscosity; multi-domain bivariate spectral quasi-linearization methods; nanofluids





## 1. Introduction

Nanofluids are advanced heat transfer fluids with applications in a large variety of industrial processes. Over the last few years, the advancements in nanofluids have generated considerable research interest on account of their novel features that make them conceivably beneficial in a number of industrial processes, such as in glass blowing, cancer therapy, plastic and polymer extrusion, micro-forming, and air conditioning.

The potential benefits and challenges of using nanofluids with controlled particle characteristics for various heat transfer applications have been investigated by many researchers. In particular, the ion-slip effects on MHD flow, which find applications in nuclear power reactors, power generation, and in several areas of astrophysics and geophysics, have been studied by [1,2]. The MHD free-forced convective nanofluid fluid flow over a permeable medium was studied by Nadeem et al. [3]. The Casson nanofluid flow with entropy generation was investigated by Haq et al. [4]. Considering mixed convection on fluid flow is very significant for industrial applications, as free or forced convection may not be enough independently to disperse adequate thermal energy. Rehman et al. [5] examined the mixed convection in a water-based nanofluid in an MHD stagnation point flow. In physical flows, a magnetic field may have a significant impact on the flow of nanofluids. Fluid flow in electrical and magnetic fields may be used, for example, to control the cooling rate. Such an implementation has been analyzed by Makinde et al. [6].

A source of heat (or sink) in a flowing fluid is increasingly becoming important to researchers. These are important in manufacturing processes and other applications, which include nuclear waste disposal, storage of food, and exothermic processes in reactors. The effect of activation energy and an exponentially fluctuating temperature-dependent source in MHD heat transfer was investigated in [7–10].

A review of previous research shows that, when estimating the performance of thermal systems, heat irreversibility is unavoidable and remains at the core of these processes. The second law of thermodynamics is used in quantifying irreversibility in optimal thermal system design. The generation of entropy is a criterion for the non-optimal operation of a thermal system. Rashidi et al. [11] studied the generation of entropy in the flow of a convective nanofluid through a vertically expanding surface. Their research found that when the Brinkman number increased, so did the level of chaos in the system. Almakki et al. [12,13] presented the formulation for entropy generation in forced convection flow of radiative viscous nanofluids. Hosseinzadeh et al. [14] calculated the entropy optimization for a magnetized flow of a nanofluid with heat radiation. Aziz et al. [15] studied the volumetric entropy production in a non-Newtonian fluid passing a stretched surface with linear thermal radiation. More current relevant research addressing the optimization of entropy in the direction of a stretched surface with varied flow parameters may be found in Pal et al. [16] and the references therein.

The aim of the current study was to investigate entropy generation and the effect of the Bejan number on 2-dimensional MHD viscoelastic incompressible nanofluid flow through an exponentially expanding sheet. The influences of dominant factors such as fluid buoyancy, viscous dissipation, heat absorption, and heat generation were studied by Maleki et al. [17]. The flow over an impermeable surface in the presence of thermal radiation and viscous dissipation was analyzed by Sharma et al. [18]. There are currently many demands for heating and cooling processes using fluids containing metallic nanoparticles. To control the rate of heat transfer in the vicinity of the expanded lamina, both entropy and Bejan number play a key role. The governing partial differential equations (PDEs) are converted to non-linear ordinary differential equations (ODEs) and solved numerically using the multi-domain bivariate spectral quasi linearization technique (MD-BSQLM). Using the MD-BSQLM approach, the linearized equations are solved numerically.

## 2. Problem Formulation

We investigated laminar free-forced convective mass-heat transportation along an exponentially stretching surface in viscous incompressible fluid in a magnetic field. We assumed that $u \to U_\infty = ax^m$ is the stretching velocity, where $m$ is the stretching rate, $T_w$ is the temperature at the surface, and $T_\infty$ is the ambient temperature, where $T_w > T_\infty$. The magnetic field is aligned in $y$-direction to influence the velocity of the fluid, as shown in Figure 1.

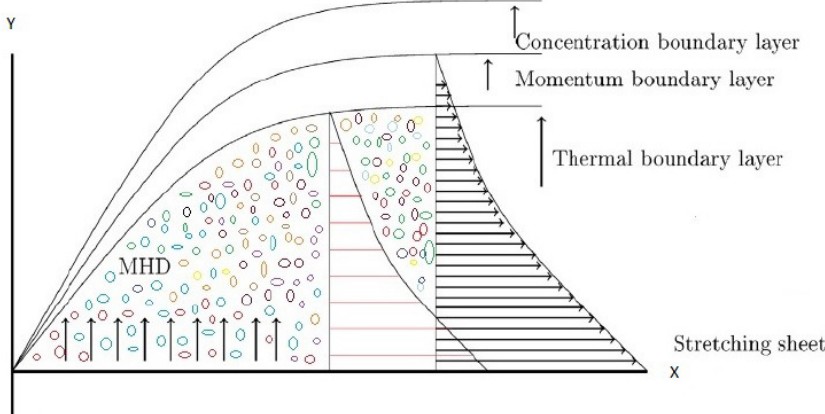

**Figure 1.** The flow schematic diagram.

The steady, incompressible fluid flow equations for a viscous nanofluid flow under these assumptions can be obtained as follows; see [19,20]:

$$\frac{\partial u}{\partial x} + \frac{\partial v}{\partial y} = 0, \tag{1}$$

$$u\frac{\partial u}{\partial x} + v\frac{\partial u}{\partial y} = \frac{1}{\rho_\infty}\frac{\partial}{\partial u}\left(\mu\frac{\partial u}{\partial y}\right) + \frac{\sigma B_0^2}{\rho_\infty}\left(U_\infty - u\right) + g\beta_T\left(T - T_\infty\right) + g\beta_C\left(C - C_\infty\right), \tag{2}$$

$$u\frac{\partial T}{\partial x} + v\frac{\partial T}{\partial y} = \alpha\frac{\partial^2 T}{\partial y^2} + \frac{Q_0}{\rho_\infty c_p}\left(T - T_\infty\right) + \frac{16\sigma^* T_\infty^3}{3k^*\rho_\infty c_p}\frac{\partial^2 T}{\partial y^2} + \frac{\nu_\infty}{c_p}\left(\frac{\partial u}{\partial y}\right)^2 + \frac{\sigma B_0^2}{\rho_\infty c_p}u^2 +$$
$$\tau\left[D_B\frac{\partial C}{\partial y}\frac{\partial T}{\partial y} + \frac{D_T}{T_\infty}\left(\frac{\partial T}{\partial y}\right)^2\right], \tag{3}$$

$$u\frac{\partial C}{\partial x} + v\frac{\partial C}{\partial y} = D_B\frac{\partial^2 C}{\partial y^2} + \frac{D_T}{T_\infty}\frac{\partial^2 T}{\partial y^2} - R_0\left(C - C_\infty\right), \tag{4}$$

where $u$ and $v$ are the velocity components in $x$ and $y$-directions, respectively; $\rho_\infty, U_\infty, T_\infty,$ $\nu_\infty,$ and $C_\infty$ represent the density, velocity, temperature, kinematic viscosity, and concentration of the fluid in free stream, respectively; $B_0$ is the external magnetic field applied in $y$-direction; $g$ is the acceleration; $\beta_C$ and $\beta_T$ are the concentration and thermal expansions, respectively; $\alpha$ represents the thermal diffusion; $Q_0$ represents the heat generation rate; $k^*$ is the mean absorption coefficient; $c_p$ is the specific heat of nanoparticles; $\sigma^*$ represents the Stefan Boltzmann constant; $\sigma$ and $\tau$ are the electrical conductivity and ratio of nanoparticle heat capacity, respectively; $D_B$ is the Brownian mention; $T$ and $C$ are the temperature and concentration distributions of the fluid, respectively; $R_0$ is a chemical reaction parameter; $D_T$ represents the thermophoretic diffusion coefficient.

The auxiliary conditions are

$$u = 0, \ vs. = 0, \ T = T_\infty, \ C = C_\infty \ \text{at} \ y = 0,$$
$$u \to U_\infty = ax^m, \ T \to T_\infty, \ C \to C_\infty \ \text{at} \ y \to \infty. \tag{5}$$

Equations (1)–(4) may be simplified using the variables [19]

$$\psi = \left(xU_\infty\nu_\infty\right)^{1/2}f(\eta,\xi), \ \eta = y\left(\frac{U_\infty}{x\nu_\infty}\right)^{1/2}, \ \xi = \frac{\sigma B_0^2}{\rho_\infty U_\infty}x,$$
$$\mu = \mu_0 e^{-\beta_1\theta(\eta,\xi)}, \ \theta(\eta,\xi) = \frac{T - T_\infty}{T_w - T_\infty}, \ h(\eta,\xi) = \frac{C - C_\infty}{C_w - C_\infty}. \tag{6}$$

Using these transformations, continuity is satisfied and Equations (2)–(4) are reduced to the following equations, where the velocity, temperature, and concentration fields are denoted by $f'(\eta,\xi), \theta(\eta,\xi),$ and $h(\eta,\xi)$, respectively:

$$f''' - \beta_1 f''\theta' + \frac{1}{2}e^{\beta_1\theta}ff'' + e^{\beta_1\theta}\left[\xi\left(1 - f'\right) + Gr_T\theta + Gr_C h\right] = \xi e^{\beta_1\theta}\left[f'\frac{\partial f'}{\partial\xi} - f''\frac{\partial f}{\partial\xi}\right], \tag{7}$$

$$\frac{1 + Nr}{Pr}\theta'' + \frac{1}{2}f\theta' + \lambda\xi\theta + Ec\left[f''^2 + \xi f'^2\right] + Nbh'\theta' + Nt\theta'^2 = \xi\left[f'\frac{\partial\theta}{\partial\xi} - \theta'\frac{\partial f}{\partial\xi}\right], \tag{8}$$

$$h'' + Le\frac{1}{2}fh' + \frac{Nt}{Nb}\theta'' - LeR_1 h = Le\xi\left[f'\frac{\partial h}{\partial\xi} - h'\frac{\partial f}{\partial\xi}\right]. \tag{9}$$

In Equations (7)–(9), the parameters are given by

$$Gr_T = \frac{g\beta_T(T_w - T_\infty)x/v_\infty^2}{U_\infty^2 v_\infty^2}, \quad Gr_C = \frac{g\beta_C(C_w - C_\infty)x/v_\infty^2}{U_\infty^2 v_\infty^2}, \quad Nr = \frac{16\sigma^* T_\infty^3}{3k^* k_f},$$

$$Pr = \frac{v_\infty}{\tau}, \quad \lambda = \frac{Q_0}{\sigma B_0^2 c_p}, \quad Ec = \frac{U_\infty}{(T_w - T_\infty)c_p},$$

$$Nb = \frac{\tau(C_w - C_\infty)D_B}{v_\infty}, \quad Le = \frac{v_\infty}{D_B}, \quad Nt = \frac{\tau(T_w - T_\infty)D_T}{T_\infty v_\infty}, \quad R_1 = \frac{R_0 x}{U_\infty},$$

where $Gr_T$ and $Gr_C$ are the Grashof numbers for temperature and concentration, respectively; $Nr$ is the thermal radiation parameter; $Nt$ is the thermophoresis parameter; $Pr$ and $Ec$ are the Prandtl and Eckert numbers; $\lambda$ is the heat generation (dimensionless); $Nb$ represents the Brownian motion parameter; and $R_1$ is the chemical reaction parameter.

The transformed boundary conditions are

$$f(0, \xi) = 0, \quad f'(0, \xi) = 0, \quad \theta(0, \xi) = 1, \quad h(0, \xi) = 1,$$
$$f'(\infty, \xi) = 1, \quad \theta(\infty, \xi) = 0, \quad h(\infty, \xi) = 0, \tag{10}$$

where the prime denotes the derivative with respect to $\eta$.

The important physical parameters for this flow with heat and mass transfer situations are the local skin-friction coefficient, the local Nusselt number, and the local Sherwood number, which can be defined as:

$$Cf_x = 2Re_x^{-1/2}f''(\xi, 0), \quad Nu_x = -Re_x^{1/2}(1 + Nr)\theta'(\xi, 0), \quad Sh_x = -Re_x^{1/2}h'(\xi, 0), \tag{11}$$

where $Re_x$ is the local Reynolds number in $x$-direction.

### 3. Entropy Generation Analysis

The principal contributors to irreversibilities in the fluid include the heat transfer through thermal radiation, viscous dissipation, magnetic field, and mass transfer, respectively.

The volumetric rate of the entropy generation rate is written as (see [21,22])

$$S'''_{gen} = \underbrace{\frac{1}{T_\infty^2}\left[k_f + \frac{16\sigma^* T_\infty^3}{3k^*}\right]\left(\frac{\partial T}{\partial y}\right)^2}_{\text{heat transfer}} + \underbrace{\frac{\mu}{T_\infty}\left(\frac{\partial u}{\partial y}\right)^2}_{\text{viscous dissipation}} + \underbrace{\frac{\sigma B_0^2}{T_\infty}u^2}_{\text{magnetic field}} +$$

$$\underbrace{\frac{RD}{C_\infty}\left[\left(\frac{\partial C}{\partial x}\right)^2 + \left(\frac{\partial C}{\partial y}\right)^2\right] + \frac{RD}{T_\infty}\left[\left(\frac{\partial T}{\partial x}\right)\left(\frac{\partial C}{\partial x}\right) + \left(\frac{\partial T}{\partial y}\right)\left(\frac{\partial C}{\partial y}\right)\right]}_{\text{mass transfer}}. \tag{12}$$

We define $S'''_0$ as the dimensionless entropy generation rate—see [12,23]—such that

$$S'''_0 = \frac{k_f\left(T_w - T_\infty\right)^2}{T_\infty^2 x^2}. \tag{13}$$

The rate of entropy generation can be expressed as

$$N_G(\eta, \xi) = \frac{S'''_{gen}}{S'''_0} = Re(1 + Nr)\theta'^2 + \frac{ReBr}{\Omega}\xi f'^2 + \Sigma\left(\frac{\chi}{\Omega}\right)^2\left[\left(Re + \frac{\eta^2}{4}\right)h'^2 - \eta\xi h'\frac{\partial h}{\partial \xi} + \right.$$

$$\left.\xi^2\left(\frac{\partial h}{\partial \xi}\right)^2\right] + \Sigma\frac{\chi}{\Omega}\left[\left(Re + \frac{\eta^2}{4}\right)\theta'h' + \xi^2\frac{\partial\theta}{\partial\xi}\frac{\partial h}{\partial\xi} - \frac{\eta\xi}{2}\left(\theta'\frac{\partial h}{\partial\xi} + h'\frac{\partial\theta}{\partial\xi}\right)\right] + \frac{ReBr}{\Omega}e^{-\beta_1\theta}f''^2. \tag{14}$$

The parameters appearing in Equation (14) are defined as

$$Re = \frac{U_\infty(x)x}{\nu_\infty}, \quad Br = \frac{\mu_\infty U_\infty^2}{k_f \Delta T}, \quad \Omega = \frac{\Delta T}{T_\infty} = \frac{T_w - T_\infty}{T_\infty},$$

$$\Sigma = \frac{C_\infty RD}{k_f}, \quad \chi = \frac{C_w - C_\infty}{C_\infty}, \tag{15}$$

where *Re* is the local Reynolds number, *Br* is the Brinkman number, $\Omega$ is the temperature difference, $\Sigma$ is the dimensionless parameter, $\chi$ and *R* are the diffusion parameter and universal gas constant, respectively, and *D* is the mass diffusion.

## 4. Method of Solution

### 4.1. Multi-Domain Bivariate Spectral Quasi-Linearization Method (MD-BSQLM)

The non-linear dimensionless ordinary differential equations were solved numerically to a high level of accuracy using the MD-BSQLM [24]. In [25,26], the method was used to solve equations describing the mixed convection flow of the power law and Casson nanofluids. We use the method to solve the nonlinear system of differential Equations (7)–(9), where we apply the multi-domain approach in the $\xi$-direction only. To use the multi-domain concept, let $\xi \in \Lambda$, where $\Lambda = [0, T]$, and consider the subdivisions

$$\Lambda_m = [\xi_{m-1}, \xi_m], \quad m = 1, 2, \cdots, p \quad \text{with} \quad 0 = \xi_0 < \xi_1 < \xi_2 < \cdots < \xi_p = T. \tag{16}$$

If the solutions to Equations (7)–(9) are denoted by $f^{(m)}(\xi_{(m)}, \eta)$, $\theta^{(m)}(\xi_{(m)}, \eta)$ and $h^{(m)}(\xi_{(m)}, \eta)$, respectively, then in the first interval $[\xi_0, \xi_1]$, the solutions

$$f^{(1)}(\xi, \eta), \quad \theta^{(1)}(\xi, \eta), \quad h^{(1)}(\xi, \eta), \tag{17}$$

are obtained subject to the "initial" conditions

$$f^{(1)}(0, \eta), \quad \theta^{(1)}(0, \eta), \quad h^{(1)}(0, \eta). \tag{18}$$

In each interval $[\xi_{m-1}, \xi_m]$, ($m \geq 2$), the continuity conditions

$$f^{(m)}(\xi_{m-1}, \eta) = f^{(m-1)}(\xi_{m-1}, \eta),$$
$$\theta^{(m)}(\xi_{m-1}, \eta) = \theta^{(m-1)}(\xi_{m-1}, \eta),$$
$$h^{(m)}(\xi_{m-1}, \eta) = h^{(m-1)}(\xi_{m-1}, \eta), \tag{19}$$

hold. This process is repeated to generate a sequence of solutions

$$f^{(m)}(\xi, \eta), \quad \theta^{(m)}(\xi, \eta), \quad h^{(m)}(\xi, \eta). \tag{20}$$

In the next step, we linearize the nonlinear system of Equations (7)–(9).

### 4.2. Linearization

The non-linear terms in Equations (7)–(9) are converted into a recursive sequence with linear components; see [27–29].

Applying the QLM technique to Equations (7)–(9) gives the following:

$$f'''^{(m)}_{e+1} + a_{1,e}f''^{(m)}_{e+1} + a_{2,e}f'^{(m)}_{e+1} + a_{3,e}f^{(m)}_{e+1} + a_{4,e}\frac{\partial f'^{(m)}_{e+1}}{\partial \xi} + a_{5,e}\frac{\partial f^{(m)}_{e+1}}{\partial \xi} + a_{6,e}\theta'^{(m)}_{e+1} + a_{7,e}\theta^{(m)}_{e+1} + a_{8,e}h^{(m)}_{e+1} = R_{1,e},$$

$$b_{0,e}\theta''^{(m)}_{e+1} + b_{1,e}\theta'^{(m)}_{e+1} + b_{2,e}\theta^{(m)}_{e+1} + b_{3,e}\frac{\partial \theta^{(m)}_{e+1}}{\partial \xi} + b_{4,e}f''^{(m)}_{e+1} + b_{5,e}f'^{(m)}_{e+1} + b_{6,e}f^{(m)}_{e+1} + b_{7,e}\frac{\partial f'^{(m)}_{e+1}}{\partial \xi} + b_{8,e}h'^{(m)}_{e+1} = R_{2,e},$$

$$h''^{(m)}_{e+1} + c_{1,e}h'^{(m)}_{e+1} + c_{2,e}h^{(m)}_{e+1} + c_{3,e}\frac{\partial h^{(m)}_{e+1}}{\partial \xi} + c_{4,e}f'^{(m)}_{e+1} + c_{5,e}f^{(m)}_{e+1} + c_{6,e}\frac{\partial f^{(m)}_{e+1}}{\partial \xi} + c_{7,e}\theta''^{(m)}_{e+1} = R_{3,e}, \qquad (21)$$

where the variable coefficients $a_{i,e}, b_{i,e}, c_{i,e}$ and $d_{i,e}(i = 1, 2, 3, \ldots)$ are presumed to be known from previous calculations and are given by

$$a_{1,e} = -\beta_1\theta'_e + \frac{1}{2}e^{\beta_1\theta_e}f_e + \xi e^{\beta_1\theta_e}\frac{\partial f_e}{\partial \xi}, \quad a_{2,e} = -\xi e^{\beta_1\theta_e}\left(1 + \frac{\partial f'_e}{\partial \xi}\right), \quad a_{3,e} = \frac{1}{2}e^{\beta_1\theta_e}f''_e, \quad a_{4,e} = -\xi e^{\beta_1\theta_e}f'_e,$$

$$a_{5,e} = \xi e^{\beta_1\theta_e}f''_e, \quad a_{6,e} = -\beta_1 f''_e, \quad a_{7,e} = \beta_1 e^{\beta_1\theta_e}\left[\frac{1}{2}f_e f''_e + \xi(1 - f') + Gr_T\theta_e + Gr_C h_e - f'_e\frac{\partial f'_e}{\partial \xi} + f''_e\frac{\partial f_e}{\partial \xi} + \frac{Gr_T}{\beta_1}\right],$$

$$a_{8,e} = e^{\beta_1\theta_e}Gr_C,$$

$$b_{0,e} = \frac{1 + Nr}{Pr}, \quad b_{1,e} = \frac{1}{2}f^{(m)}_e + N_b h'^{(m)}_e + 2N_t\theta'^{(m)}_e + \xi\frac{\partial f^{(m)}_e}{\partial \xi}, \quad b_{2,e} = \lambda\xi, \quad b_{3,e} = -\xi f'^{(m)}_e, \quad b_{4,e} = 2Ecf''^{(m)}_e,$$

$$b_{5,e} = 2\xi Ecf'^{(m)}_e - \xi\frac{\partial \theta^{(m)}_e}{\partial \xi}, \quad b_{6,e} = \frac{1}{2}\theta'^{(m)}_e, \quad b_{7,e} = \xi\theta'^{(m)}_e, \quad b_{8,e} = N_b\theta'^{(m)}_e,$$

$$c_{1,e} = \frac{1}{2}Lef^{(m)}_e + Le\xi\frac{\partial f^{(m)}_e}{\partial \xi}, \quad c_{2,e} = -LeR_1, \quad c_{3,e} = -Le\xi f'^{(m)}_e,$$

$$c_{4,e} = -Le\xi\frac{\partial h^{(m)}_e}{\partial \xi}, \quad c_{5,e} = \frac{1}{2}Leh'^{(m)}_e, \quad c_{6,e} = Le\xi h'^{(m)}_e, \quad c_{7,e} = \frac{N_t}{N_b}.$$

Here, the right-hand side terms $R_{i,e}(i = 1, 2, 3)$ in Equation (21) are given below:

$$R_{1,e} = f'''^{(m)}_e + a_{1,e}f''^{(m)}_e + a_{2,e}f'^{(m)}_e + a_{3,e}f^{(m)}_e + a_{4,e}\frac{\partial f'^{(m)}_e}{\partial \xi} + a_{5,e}\frac{\partial f^{(m)}_e}{\partial \xi} + a_{6,e}\theta'^{(m)}_e + a_{7,e}\theta^{(m)}_e + a_{8,e}h^{(m)}_e - F_f,$$

$$R_{2,e} = b_{0,e}\theta''^{(m)}_e + b_{1,e}\theta'^{(m)}_e + b_{2,e}\theta^{(m)}_e + b_{3,e}\frac{\partial \theta^{(m)}_e}{\partial \xi} + b_{4,e}f''^{(m)}_e + b_{5,e}f'^{(m)}_e + b_{6,e}f^{(m)}_e + b_{7,e}\frac{\partial f^{(m)}_e}{\partial \xi} + b_{8,e}h'^{(m)}_e - F_\theta,$$

$$R_{3,e} = h''^{(m)}_e + c_{1,e}h'^{(m)}_e + c_{2,e}h^{(m)}_e + c_{3,e}\frac{\partial h^{(m)}_e}{\partial \xi} + c_{4,e}f'^{(m)}_e + c_{5,e}f^{(m)}_e + c_{6,e}\frac{\partial f^{(m)}_e}{\partial \xi} + c_{7,e}\theta''^{(m)}_e - F_h, \qquad (22)$$

and the functions $F_f, F_\theta$, and $F_h$ are as defined in Equations (7)–(9).

### 4.3. Collocation

The interval $\xi \in \Lambda_m \equiv [\xi_{m-1}, \xi_m]$ is transformed to $s \in [-1, 1]$ using the linear transformation

$$\xi = \frac{1}{2}(\xi_m - \xi_{m-1})s + \frac{1}{2}(\xi_m + \xi_{m-1}),$$

and $\eta \in [0, \infty]$ is transformed to $x \in [-1, 1]$ using

$$\eta = \frac{1}{2}L(x + 1)$$

where $L$ is a number large enough to represent the condition at infinity. For collocation, we use the Gauss–Lobatto collocation points defined as

$$x_i = \cos\left(\frac{\pi i}{N_x}\right), \qquad s_j = \cos\left(\frac{\pi j}{N_s}\right), \qquad i = 0, 1, \ldots, N_x, \qquad j = 0, 1, \ldots, N_s.$$

The Chebyshev differentiation matrix is as defined in [30], with respect to $\zeta$. The general form can be expressed as

$$\left.\frac{\partial^n f^{(m)}}{\partial x^n}\right|_{(s_j, x_i)} = \mathbf{D}^n F_i^{(m)}, \qquad \left.\frac{\partial f^{(m)}}{\partial s}\right|_{(s_j, x_i)} = \sum_{q=0}^{Ns} \mathbf{d}_{iq}^{(m)} F_q^{(m)},$$

$$\left.\frac{\partial^n \theta^{(m)}}{\partial x^n}\right|_{(s_j, x_i)} = \mathbf{D}^n T_i^{(m)}, \qquad \left.\frac{\partial \theta^{(m)}}{\partial s}\right|_{(s_j, x_i)} = \sum_{q=0}^{Ns} \mathbf{d}_{iq}^{(m)} T_q^{(m)},$$

$$\left.\frac{\partial^n h^{(m)}}{\partial x^n}\right|_{(s_j, x_i)} = \mathbf{D}^n H_i^{(m)}, \qquad \left.\frac{\partial h^{(m)}}{\partial s}\right|_{(s_j, x_i)} = \sum_{q=0}^{Ns} \mathbf{d}_{iq}^{(m)} H_q^{(m)}, \qquad i = 0, 1, \ldots, Nx, \text{ (23)}$$

where

$$F_i = [f(s_j, x_0), f(s_j, x_1), \ldots, f(s_j, x_{Nx})]^T, \quad T_i = [\theta(s_j, x_0), \theta(s_j, x_1), \ldots, \theta(s_j, x_{Nx})]^T,$$

$$H_i = [h(s_j, x_0), h(s_j, x_1), \ldots, h(s_j, x_{Nx})]^T, \quad \mathbf{d} = \frac{2d}{\zeta_m - \zeta_{m-1}}, \quad \mathbf{D} = \frac{2D}{C}.$$

By applying Equations (21)–(23), we obtain

$$\left(\mathbf{D}^3 + \mathbf{a_{1,e}}\mathbf{D}^2 + \mathbf{a_{2,e}}\mathbf{D} + \mathbf{a_{3,e}}I + \mathbf{a_{4,e}}\mathbf{d}_{ii}^{(m)}(\mathbf{D}) + \mathbf{a_{5,e}}\mathbf{d}_{ii}^{(m)}I\right)F_{i,e+1}^{(m)}$$

$$+ \left(\mathbf{a_{4,e}}\sum_{\substack{q=0 \\ q\neq i}}^{Ns-1} \mathbf{d}_{iq}^{(m)}(\mathbf{D}) + \mathbf{a_{5,e}}\sum_{\substack{q=0 \\ q\neq i}}^{Ns-1} \mathbf{d}_{iq}^{(m)}I\right)F_{q,e+1}^{(m)} + \left(\mathbf{a_{6,e}}\mathbf{D} + \mathbf{a_{7,e}}I\right)T_{i,e+1}^{(m)}$$

$$+ \mathbf{a_{8,e}}H_{i,e+1}^{(m)} = RR_{1,e},$$

$$\left(b_{0,e}\mathbf{D}^2 + \mathbf{b_{1,e}}\mathbf{D} + b_{2,e}I + \mathbf{b_{3,e}}\mathbf{d}_{ii}^{(m)}\right)T_{i,e+1}^{(m)} + \left(\mathbf{b_{3,e}}\sum_{\substack{q=0 \\ q\neq i}}^{Ns-1} \mathbf{d}_{iq}^{(m)}I\right)T_{q,e+1}^{(m)}$$

$$+ \left(\mathbf{b_{4,e}}\mathbf{D}^2 + \mathbf{b_{5,e}}\mathbf{D} + \mathbf{b_{6,e}}I + \mathbf{b_{7,e}}\mathbf{d}_{ii}^{(m)}I\right)F_{i,e+1}^{(m)} + \left(\mathbf{b_{7,e}}\sum_{\substack{q=0 \\ q\neq i}}^{Ns-1} \mathbf{d}_{iq}^{(m)}I\right)F_{q,e+1}^{(m)}$$

$$+ \mathbf{b_{8,e}}\mathbf{D}P_{i,e+1}^{(m)} = RR_{2,e},$$

$$\left(\mathbf{D}^2 + \mathbf{c_{1,e}}\mathbf{D} + c_{2,e}I + \mathbf{c_{3,e}}\mathbf{d}_{ii}^{(m)}\right)H_{i,e+1}^{(m)} + \left(\mathbf{c_{3,e}}\sum_{\substack{q=0 \\ q\neq i}}^{Ns-1} \mathbf{d}_{iq}^{(m)}I\right)H_{q,e+1}^{(m)}$$

$$+ \left(\mathbf{c_{4,e}}\mathbf{D} + \mathbf{c_{5,e}}I + \mathbf{c_{6,e}}\mathbf{d}_{ii}^{(m)}I\right)F_{i,e+1}^{(m)} + \left(\mathbf{c_{6,e}}\sum_{\substack{q=0 \\ q\neq i}}^{Ns-1} \mathbf{d}_{iq}^{(m)}I\right)F_{q,e+1}^{(m)}$$

$$+ c_{7,e}\mathbf{D}^2 T_{i,e+1}^{(m)} = RR_{3,e}, \tag{24}$$

where the bold variable coefficients represent diagonal matrices

$$RR_{1,e} = R_{1,e} - \left( \mathbf{a_{4,e}} \mathbf{d}_{iNs}^{(m)} (\mathbf{D}) + \mathbf{a_{5,e}} \mathbf{d}_{iNs}^{(m)} I \right) F_{N_s,e}^{(m)},$$
$$RR_{2,e} = R_{2,e} - \mathbf{b_{3,e}} \mathbf{d}_{iNs}^{(m)} T_{N_s,r}^{(m)} - \mathbf{b_{7,e}} \mathbf{d}_{iNs}^{(m)} F_{N_s,e}^{(m)}, \quad \text{and}$$
$$RR_{3,e} = R_{3,e} - \mathbf{c_{3,e}} \mathbf{d}_{iNs}^{(m)} H_{N_s,r}^{(m)} - \mathbf{c_{6,e}} \mathbf{d}_{iNs}^{(m)} F_{N_s,e}^{(m)}, \tag{25}$$

and I is an $(Nx + 1) \times (Nx + 1)$ identity matrix. The equation is then expressed in matrix form as

$$\begin{bmatrix} A11_{i,j} & A12_{i,j} & A13_{i,j} \\ A21_{i,j} & A22_{i,j} & A23_{i,j} \\ A31_{i,j} & A32_{i,j} & A33_{i,j} \end{bmatrix} \begin{bmatrix} F_i^{(m)} \\ T_i^{(m)} \\ P_i^{(m)} \end{bmatrix} = \begin{bmatrix} RR_{1,e}^{(m)} \\ RR_{2,e}^{(m)} \\ RR_{3,e}^{(m)} \end{bmatrix} \tag{26}$$

where

$$A11_{i,i} = \mathbf{D}^3 + \mathbf{a_{1,e}} \mathbf{D}^2 + \mathbf{a_{2,e}} \mathbf{D} + \mathbf{a_{3,e}} I + \mathbf{a_{4,e}} \mathbf{d}_{ii}^{(m)} (\mathbf{D}) + \mathbf{a_{5,e}} \mathbf{d}_{ii}^{(m)} I, \quad A11_{i,j} = \mathbf{a_{4,e}} \mathbf{d}_{ij}^{(m)} (\mathbf{D}) + \mathbf{a_{5,e}} \mathbf{d}_{ij}^{(m)} I$$
$$A12_{i,i} = \mathbf{a_{6,e}} \mathbf{D} + \mathbf{a_{7,e}} I, \qquad A13_{i,i} = a_{8,e} I, \qquad A12_{i,j} = A13_{i,j} = A23_{i,j} = A32_{i,j} = \mathbf{0}$$
$$A21_{i,i} = \mathbf{b_{4,e}} \mathbf{D}^2 + \mathbf{b_{5,e}} \mathbf{D} + \mathbf{b_{6,e}} I + \mathbf{b_{7,e}} \mathbf{d}_{ii}^{(m)} I, \qquad A21_{i,j} = \mathbf{b_{7,e}} \mathbf{d}_{ij}^{(m)} I$$
$$A22_{i,i} = b_{0,e} \mathbf{D}^2 + \mathbf{b_{1,e}} \mathbf{D} + b_{2,e} I + \mathbf{b_{3,e}} \mathbf{d}_{ii}^{(m)} I \qquad A22_{i,j} = \mathbf{b_{3,e}} \mathbf{d}_{ij}^{(m)} I, \qquad A23_{i,i} = \mathbf{b_{8,e}} \mathbf{D},$$
$$A31_{i,i} = \mathbf{c_{4,e}} \mathbf{D} + \mathbf{c_{5,e}} I + \mathbf{c_{6,e}} \mathbf{d}_{ii}^{(m)} I, \qquad A31_{i,j} = \mathbf{c_{6,e}} \mathbf{d}_{ij}^{(m)} I \qquad A32_{i,i} = c_{7,e} \mathbf{D}^2,$$
$$A33_{i,i} = \mathbf{D}^2 + \mathbf{c_{1,e}} \mathbf{D} + c_{2,e} I + \mathbf{c_{3,e}} \mathbf{d}_{ii}^{(m)} I \qquad A33_{i,j} = \mathbf{c_{3,e}} \mathbf{d}_{ij}^{(m)} I, \tag{27}$$

Here, $A_{rs}(i,i)$ is the diagonal of each matrix $A_{rs}(i,j)$, where $r = s = 1, 2, 3$.

## 5. Convergence Analysis

The convergence of the MD-BSQLM is discussed in this section. The convergence rate is determined using the infinity norm of the error given by:

$$|| Q_{s+1} ||_\infty = \max_{0 \le \varepsilon \le N_x} |Q_{s+1} - Q_s| \quad \text{for } Q = (f, \theta, h). \tag{28}$$

The differences between the approximate values of the functions at the current and previous iterations are depicted in Figures 2 and 3. The convergence of the method is achieved by the sixth iteration for both the functions and their derivatives. The graphs show that the MD-BSQLM converges rapidly with a high degree of accuracy. The method may be extended to solve other types of highly non-linear differential equations.

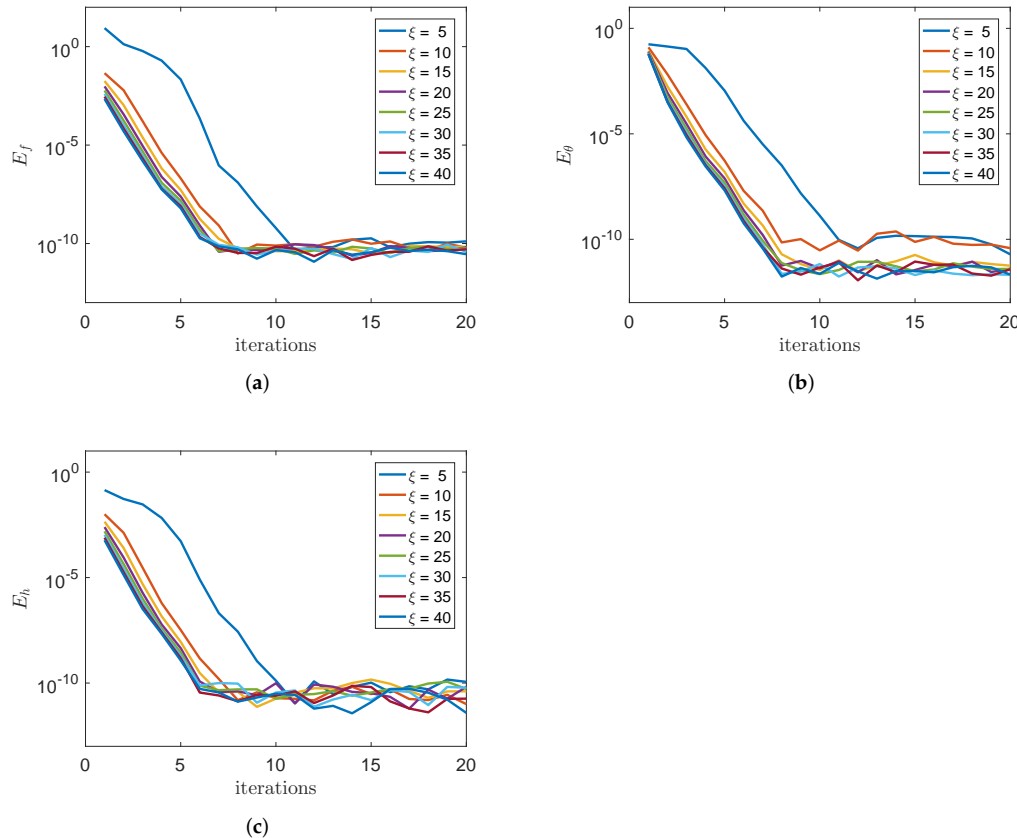

**Figure 2.** Error in (**a**) $f(\eta, \xi)$ (**b**) $\theta(\eta, \xi)$, and (**c**) $h(\eta, \xi)$ for different values of $\xi$.

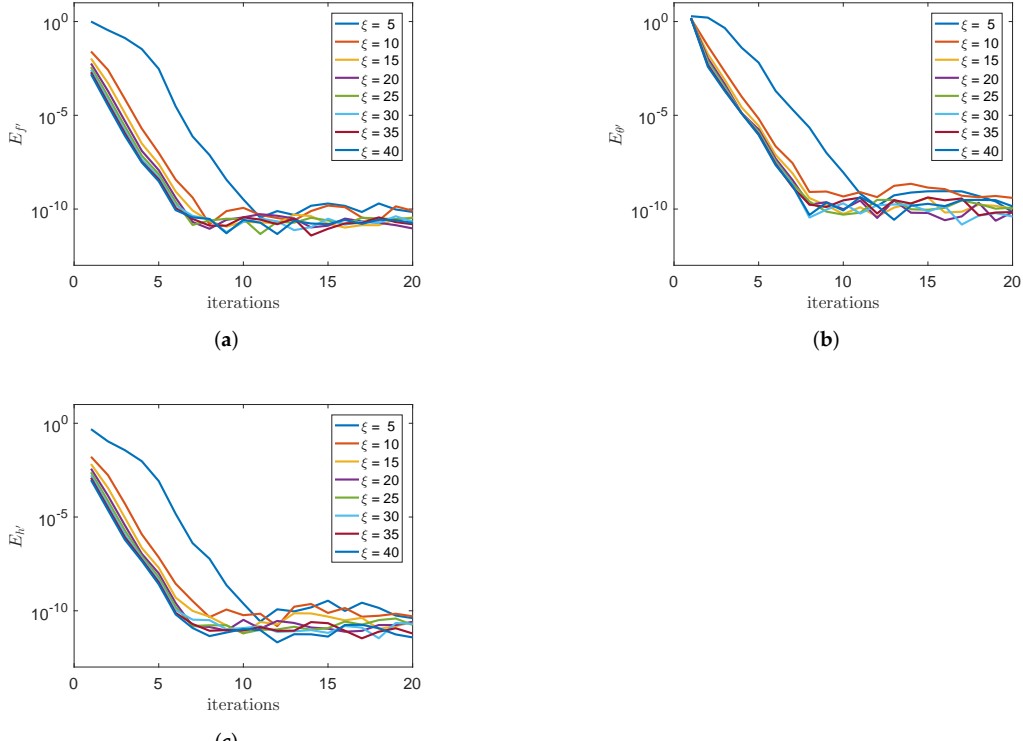

**Figure 3.** Error in (**a**) $f'(\eta, \xi)$ (**b**) $\theta'(\eta, \xi)$, and (**c**) $h'(\eta, \xi)$ for different values of $\xi$.

## 6. Results and Discussion

In this section, we present tabulated results and graphical representations of different features of the fluid flow and heat and mass transfer under the effects of several parameters. The quantities of engineering interest, such as skin fraction and Nusselt and Sherwood numbers are computed and presented in Table 1. The fluid flow equations were solved numerically using the multi-domain bivariate spectral quasi-linearization method for the chosen parameter values.

**Table 1.** Skin friction, and heat and mass transfer coefficients for $\beta_1 = 0.5, Gr_T = Gr_C = 0.5$, $\lambda = -0.09, Ec = 0.05, Le = 0.4$, and $R_1 = 0.3$.

| $\xi$ | $Nt$ | $Nb$ | $Nr$ | $Pr$ | $Cf_x$ | $Nu_x$ | $Sh_x$ |
|-------|------|------|------|------|--------|--------|--------|
| 5 | | | | | 3.0874257 | −1.5088474 | 0.4653439 |
| 10 | 0.001 | 0.001 | 0.5 | 6.8 | 4.3131008 | −1.9976737 | 0.9329592 |
| 15 | | | | | 5.2604334 | −2.3949035 | 1.3198495 |
| 20 | | | | | 6.0620646 | −2.7411796 | 1.6592534 |
| | 0.001 | | | | 5.187825 | 2.154772 | −0.008580 |
| 10 | 0.3 | 0.001 | 0.5 | 6.8 | 5.488176 | 1.989699 | −3.770309 |
| | 0.5 | | | | 5.656798 | 1.881303 | −5.678613 |
| | 0.7 | | | | 5.803213 | 1.776072 | −7.140323 |
| | | 0.001 | | | 6.003369 | 2.503334 | −0.068861 |
| 15 | 0.001 | 0.1 | 0.5 | 6.8 | 5.992718 | 2.466485 | −0.033776 |
| | | 0.15 | | | 5.988566 | 2.520528 | −0.027644 |
| | | 0.2 | | | 5.970386 | 3.233199 | −0.013723 |
| | | | 0.001 | | 5.198892 | 2.432598 | −0.011509 |
| 10 | 0.001 | 0.001 | 0.1 | 6.8 | 5.196294 | 2.362772 | −0.010779 |
| | | | 0.2 | | 5.193907 | 2.301212 | −0.010132 |
| | | | 0.3 | | 5.191718 | 2.246864 | −0.009559 |

A comparison with previously computed results is depicted in Table 2. These results are comparable to those of Chamkha et al. [19] and Yih [31], which show that this method is robust and gives accurate results.

**Table 2.** Comparison of the values of $-\theta'(0,0)$ for different values of $Pr$.

| $Pr$ | Chamkha et al. [19] Finite-Difference Method | Yih [31] Keller Box Method | Present Results MD-BSQLM |
|------|---------------------------------------------|---------------------------|-------------------------|
| 0.1 | 0.142003 | 0.140034 | 0.1400343 |
| 1.0 | 0.332173 | 0.332057 | 0.3320571 |
| 10.0 | 0.728310 | 0.728141 | 0.7281412 |
| 100.0 | 1.572180 | 1.571831 | 1.5718323 |
| 1000.0 | 3.388090 | 3.387083 | 3.3870854 |
| 10,000.0 | 7.300800 | 7.297402 | 7.2974001 |

Figure 4a,c show increases in velocity and concentration with higher values of the variable viscosity parameter ($\beta_1$). This is true from the Poiseuille law, as the pressure and viscosity within the flow region increase, leading to the observed increase in both flow rate and concentration. The fluid temperature reduces, as most of the thermal energy is lost through convection and conduction to the walls of the sheet, as depicted in Figure 4b.

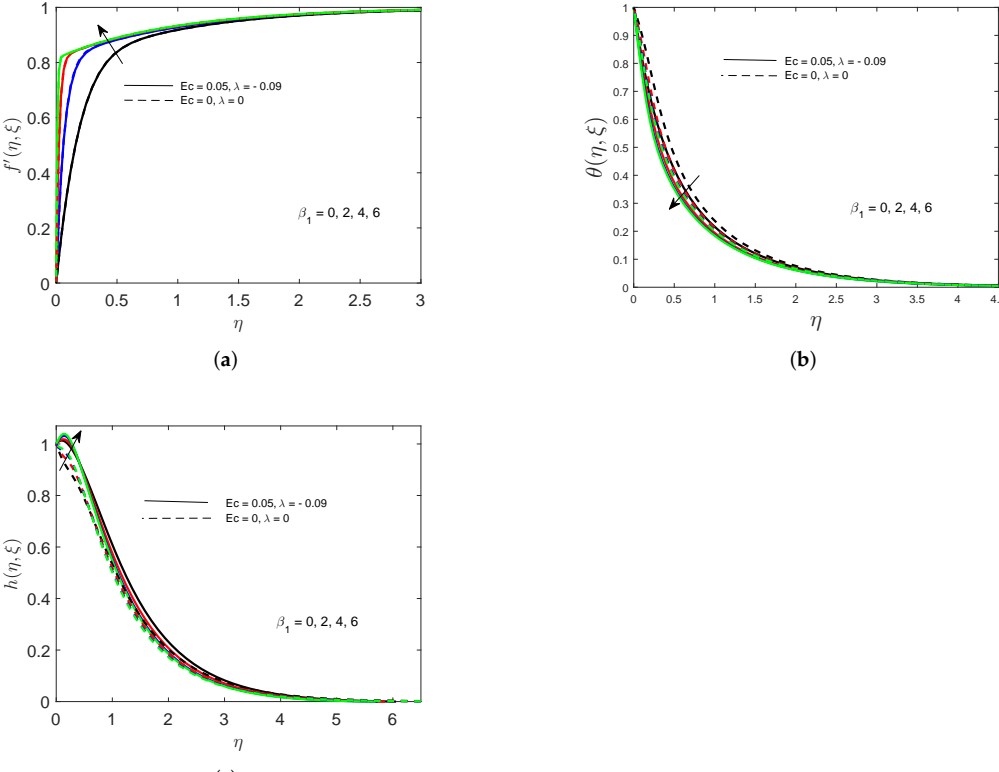

**Figure 4.** (**a**) Velocity; (**b**) temperature, and (**c**) concentration profiles for different values of the viscosity parameter.

Table 1 gives the computed skin friction and heat and mass transfer coefficients for different values of the selected parameters. It is noted that the coefficient of skin friction increases for higher values of the non-dimensional number ($\xi$), thermal radiation ($Nr$), and thermophoresis parameters ($Nt$). The dynamic pressure on the surface of the sheet is increased due to the force generated by an increased temperature gradient. The heat transfer coefficient was observed to decrease with higher values of $Nr$ and $Nt$ and increase with $\xi$. The exponentially stretching sheet of the vibrating sheet emits thermal energy through conduction of the fluid particles, which causes the Nusselt number to decrease. The Sherwood number increases with higher values of $\xi$ and $Nt$ and decreases with an increase in the Brownian motion ($Nb$) or any thermophoresis parameter. The force generated by the increased temperature gradient in the fluid causes an increase in mass energy transfer through convection at the vibrating sheets, resulting in a higher mass transfer coefficient. Similar results were obtained by [32,33].

Higher emitted thermal radiation increases the boundary layer temperature, as clearly depicted in Figure 5a. Heat energy in the form of electromagnetic radiation is emitted by the vibrating sheets and transferred to the fluid, causing the temperature to increase. The concentration profiles decrease with higher values of $Nr$, as displayed in Figure 5b.

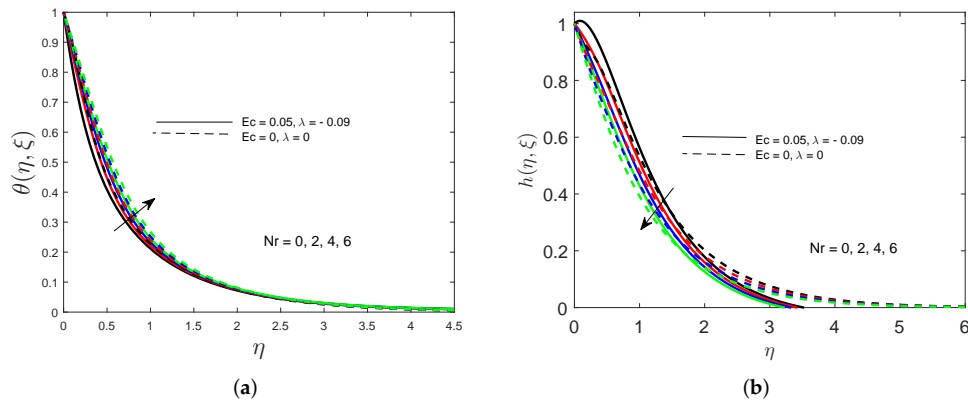

**Figure 5.** The impacts of thermal radiation on (**a**) temperature and (**b**) concentration profiles.

The Soret effect is observed in Figure 6a–c. Higher values of the thermophoresis parameter lead to increases in all three flow profiles. This can be explained by the fact that the exponentially stretching sheet causes an upward shift in the temperature gradient. The movement of fluid particles in the hot region and high energy levels displace the fluid particles in the cold region, making the three flow variables increase. Similar results were obtained by [34].

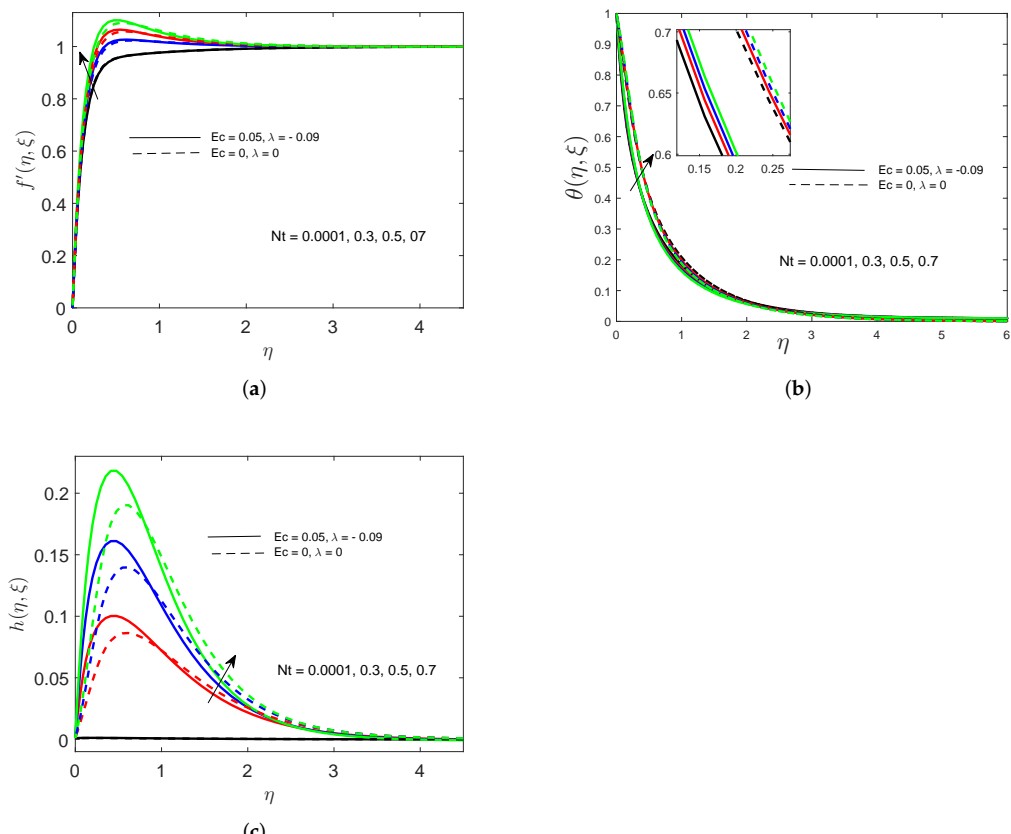

**Figure 6.** Impacts of the thermophoresis parameter on (**a**) velocity, (**b**) temperature, and (**c**) concentration profiles.

The impacts of the local Reynolds (*Re*) and Brinkman numbers (*Br*) on entropy generation are displayed in Figure 7a,b, respectively. Entropy generation increases with higher values of the Reynolds number. The force due to the momentum of the fluid flow causes the inertial forces to increase, and as a result, the irreversibility associated with the flow

increases. A similar trend can be observed for the Brinkman number. With higher values of *Br*, more thermal energy is produced by viscous dissipation than thermal energy produced by molecular dissipation. This will thus mean slower heat conduction, and the temperature rises, causing more entropy generation.

The influences of the temperature difference parameter ($\Omega$) and the temperature Grashof number ($Gr_T$) on entropy generation are depicted in Figure 8a,b, respectively. Entropy generation can be observed to reduce with higher values of $\Omega$; the opposite can be observed with increasing values of $Gr_T$. The rate of entropy generation increases as a result of increased buoyancy force due to spatial variation in the density of the fluid caused by the temperature difference.

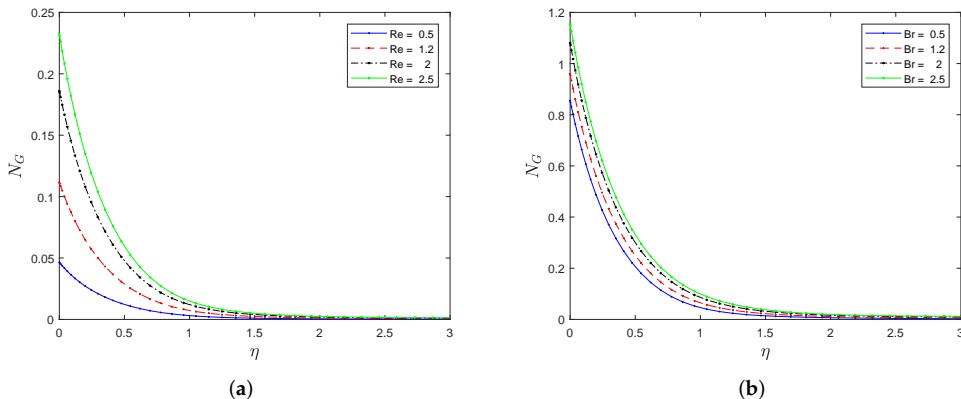

(**a**)            (**b**)

**Figure 7.** The influences of (**a**) the local Reynolds and (**b**) Brinkman numbers on the entropy generation profiles.

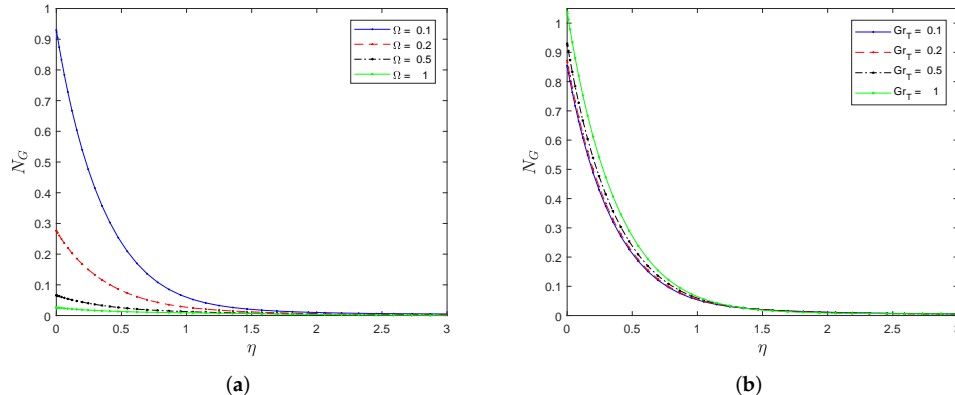

(**a**)            (**b**)

**Figure 8.** The influences of (**a**) the temperature difference parameter, and (**b**) temperature Grashof number on the entropy generation profiles.

## 7. Conclusions

In this study, the physical influences of certain parameters on nanofluid flow and heat-mass transfer past an exponentially stretching sheet, and entropy generation, have been analyzed. From this study, we may conclude as follows:

1.  The MD-BSQLM converges rapidly with a high degree of accuracy. The accuracy may be enhanced by increasing the number of collocation points.
2.  Increases in thermophoresis and thermal radiation parameters lead to increases in both the skin friction and mass transfer coefficients.
3.  An increase in thermophoresis, Brownian motion, or thermal radiation parameters leads to a decrease in the rate of heat transfer.
4.  The entropy generation rate can be minimized through the temperature difference parameter and temperature Grashof number.

**Author Contributions:** Formal analysis, M.A.; Methodology, H.M.; Resources, Z.M.; Supervision, P.S. All authors have read and agreed to the published version of the manuscript.

**Funding:** This research received no external funding.

**Institutional Review Board Statement:** Not applicable.

**Informed Consent Statement:** Not applicable.

**Data Availability Statement:** Not applicable.

**Conflicts of Interest:** The authors declare no conflict of interest.

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
