# Peer review of "Entropy Optimization in MHD Nanofluid Flow over an Exponential Stretching Sheet"

_applsci, doi:10.3390/app122110809_

Round 1

Reviewer 2 Report

The behavior of mixed convective nanofluid flow in an exponential stretching sheet with temperature-dependent viscosity is examined by the authors. The boundary layer flow's governing equations were converted into highly nonlinear ordinary differential equations using the similarity technique. The equations are resolved using the multidomain spectral quasi-linearization technique.

It can be accepted with the following suggestions.

1.      Add some more results in an abstract.

2.      Provide reference support for neglecting viscous dissipation.

3.      Reconfirm the arrangement of boundary layer notation in Fig. 1. It is either momentum, thermal and concentration or thermal, momentum and concentration?

4.      Reconfirm all mathematical equations and add reference support where needed.

5.      Improve the literature survey by following :

Appl. Sci. 202212(5), 2383; https://doi.org/10.3390/app12052383,

Appl. Sci. 202212(4), 2000; https://doi.org/10.3390/app12042000

Round 2

Reviewer 2 Report

Accepted